# Impact of health insurance coverage for *Helicobacter pylori* gastritis on the trends in eradication therapy in Japan: retrospective observational study and simulation study based on real-world data

Shinzo Hiroi,[1,2] Kentaro Sugano,[3] Shiro Tanaka,[4] Koji Kawakami[1]

[1]Department of Pharmacoepidemiology, Graduate School of Medicine and Public Health, Kyoto University, Kyoto, Japan
[2]Japan Medical Affairs, Takeda Pharmaceutical Company Limited, Tokyo, Japan
[3]Department of Medicine, Jichi Medical University, Tochigi, Japan
[4]Department of Clinical Biostatistics, Graduate School of Medicine, Kyoto University, Kyoto, Japan

**Correspondence to**
Dr Koji Kawakami;
kawakami.koji.4e@kyoto-u.ac.jp

## ABSTRACT

**Objectives** To explore the prevalence of *Helicobacter pylori* infection in Japan and the trends of its eradication therapy before and after the changes of the insurance coverage policy, first started in 2000, and expanded to cover *H. pylori*-positive gastritis in 2013. The impacts that the changes brought were estimated.

**Methods** In this retrospective observational study and simulation study based on health insurance claims data, product sales data and relevant studies, individuals who received triple therapy (amoxicillin, clarithromycin, proton-pump inhibitors or potassium-competitive acid blockers) were defined as the first-time patients for *H. pylori* eradication in two Japanese health insurance claims databases (from approximately 1.6 million and 10.5 million individuals). Each sales data of eradication packages and examination kits were used to estimate the number of *H. pylori*-eradicated individuals nationwide. The prevalence of *H. pylori* infection, including the future rate, was predicted using previous studies and the estimated population trend by a national institute. Cases completed prior to the policy change on insurance coverage were simulated to estimate what would have happened had there been no change in the policy.

**Results** The numbers of patients first received eradication therapy were 81 119 and 170 993 from two databases. The nationwide estimated number of patients successfully eradicated was approximately 650 000 per year between 2001 and 2012, whereas it rapidly rose to 1 380-000 per year in 2013. The estimated prevalence of infection in 2050 is 5%, this rate was estimated to be 28% and 22% if the policy changes had not occurred in 2000 and 2013, respectively.

**Conclusions** The impact of policy changes for *H. pylori* eradication therapy on the prevalence of infection was shown. The results suggest that insurance coverage expansion may also reduce the prevalence in other countries with a high prevalence of *H. pylori* infection if the reinfection is low.

## Strengths and limitations of this study

► Demonstrates for the first time the impact of insurance policy expansion for *Helicobacter pylori* eradication therapy in a quantitative manner based on an analysis of nationwide real-world data.
► Robust and reliable results were obtained from combinations of large-scale insurance claims databases and sales data of the most commonly used eradication treatments and test kits.
► The success rate of eradication was obtained from previous studies; therefore, the rate might be different from current clinical practice.
► The health insurance claims databases have potential biases: in one database, the information on individuals older than 65 years is limited because it is the information from employed individuals and their family members, whereas another database included the data only from large hospitals.

## INTRODUCTION

Throughout the world, gastric cancer is one of the most common cancers; 952 000 new patients were diagnosed in 2012.[1] The incidence of gastric cancer is higher in Asian countries; Korea, Japan and China have the first, third and fifth highest rates, respectively, in the world.[2] In Japan, the prevalence and mortality of gastric cancer are constantly among the top three of all cancers. Therefore, it is considered to be one of the highest priorities in preventive policy. *Helicobacter pylori* can cause gastric inflammation, which can then lead to gastric and duodenal ulcers, as well as gastric cancer.[3–5] Thus, eradication of *H. pylori* is considered as an effective therapy in reducing the risk of those diseases.

Due to the concern of high gastric cancer prevalence in East Asian countries, some

preventive programmes have been launched to reduce the incidence of gastric cancer. In Korea, a cancer screening programme was established by the government to provide for almost all people of eligible age (40 years or older for gastric cancer) with free screening or provision at minimum cost in 1999.[6] Large clinical trials and health economic studies have been conducted in China, and a consensus statement was formulated to encourage *H. pylori* eradication therapy.[6 7] In Taiwan, the results of a community-level large screening and eradication programme, as well as a health economic evaluation, support the efficacy of *H. pylori* eradication therapy.[8] In Japan, in November 2000, based on the results of diverse clinical studies,[3 9–20] the government approved the addition of *H. pylori* eradication therapy in their insurance policy as a treatment for *H. pylori*-positive gastric ulcer and duodenal ulcer. Furthermore, insurance coverage was expanded in June 2010 to include gastric mucosa-associated lymphoid tissue lymphoma, idiopathic thrombocytopenic purpura and postendoscopic resection of early gastric cancer, and in February 2013, to include *H. pylori*-positive gastritis based on the recommendations of Japanese guideline.[21]

Japanese health insurance is a system of universal coverage; the effect of change in health insurance coverage policy is spread throughout the nation. In terms of *H. pylori* eradication, anyone diagnosed with a disease covered for eradication therapy by health insurance can receive eradication therapy with coverage. Therefore, health insurance reimbursement seems to have the same or greater impact on clinical practice as recommendations from diagnostic/treatment guidelines in countries where universal health insurance coverage is established, such as in Japan and Korea. Various sizes of preventative programmes for gastric cancer have been implemented in the high prevalence countries for both gastric cancer and *H. pylori* infection. In some countries, *H. pylori* eradication therapy for patients with *H. pylori*-positive gastric ulcer and duodenal ulcer has been covered by national health insurance. However, eradication therapy for *H. pylori*-positive gastritis has not been covered to date in these countries other than Japan.[8] The effect of insurance coverage expansion on the prevalence of *H. pylori* infection has been evaluated in only a few studies at the community level in Japan.[22 23] Nonetheless, the national-level prevalence rate of *H. pylori* infection has not been reported and its change has not been assessed after the insurance coverage for *H. pylori* eradication therapy was expanded to include *H. pylori*-positive gastritis in 2013. The progressive insurance expansion was reported to be efficient,[24] and the incidence of peptic ulcer has decreased since the change in insurance coverage policy for *H. pylori* eradication in 2000.[25] They also estimated that gastric cancer mortality would decrease based on the assumption that 50% of patients infected with *H. pylori* would receive eradication therapy.[26] However, this estimate was based on neither the observed number of patients undergoing *H. pylori* eradication nor the prevalence rate of infection. To evaluate the impact of changes

to the insurance policy on the incidence of various diseases, including gastric cancer, it is necessary to elucidate the national trend of eradication therapy and the prevalence rate of infection before and after the changes in the insurance policy.

The primary objective of this study was to assess how health insurance policy changes have impacted eradication therapy and the prevalence rate of *H. pylori* infection in Japan. Furthermore, the future effect, as a result of the policy changes, on the prevalence of *H. pylori* infection was evaluated. In this study, health insurance claims databases and product sales data were used to estimate the number of eradication treatments. The successful eradication rate from 2000 onwards, at which time the health insurance began its coverage for the eradication therapy, was also estimated. The prevalence rate of infection and the number of infected individuals up to 2060 were predicted as based on the above data analysis and the prevalence rate of *H. pylori* infection as reported in previous studies. Furthermore, a simulation was conducted to estimate what the probable effects would be had the policy changes not been made.

## METHODS
### Data sources
Insurance claims databases from Japan Medical Data Center (JMDC) from January 2005 to December 2015 and Medical Data Vision (MDV) from April 2008 to December 2015 were used for the analyses. The JMDC database is a registry of health insurance claims and medical examination records for insured individuals and their families in more than 50 health insurance societies. Because this database only included information on company employees and their families, the information for those older than 65 years was limited. Also, there were no data for those older than 75 years. Until 2014, this database covered 1.6 million individuals which accounted for 1% of the Japanese population. The medical database from MDV covered 10.5 million individuals in 192 acute care hospitals using diagnostic procedure combination/per diem payment system (DPC/PDPS). It included 11% of acute care hospitals in Japan with the number of beds from 20 to more than 1000. These databases included the patient's gender, age, diagnosis, prescription information and so on. Diagnosis information is based on the International Classification of Diseases, tenth revision, and drugs are coded in the Anatomical Therapeutic Chemical Classification System. Both databases included anonymous and personally unidentifiable data.

To estimate the nationwide number of infected individuals, product sales data for the most common eradication medicine and test kit for *H. pylori* infection were analysed. The sales data for eradication medicine, Lansap (Takeda Pharmaceutical), which consists of lansoprazole, amoxicillin and clarithromycin in one package, was provided by the manufacturer from December 2002 to December 2015. The data for $^{13}$C-urea breath test (UBIT, Otsuka

**Table 1** Studies on the number of *Helicobacter pylori*-infected individuals and prevalence rate of infection

| First author | No of subjects | Population | Study design | Observation year |
|---|---|---|---|---|
| Asaka[27] | 426 | Asymptomatic children, students and adults (participating at the health screening centre) living in Sapporo, Hokkaido | Observational study | 1990 |
| Fujisawa[31] | 349 | Healthy persons living in seven prefectures in the central part of Japan | Observational study | 1974 |
| | 324 | | | 1984 |
| | 342 | | | 1994 |
| Watabe[32] | 6 983 | Participants in a mass health appraisal programme | Observational study | 1996 |
| Ueda[33] | 14 716 | Individuals who underwent a health check-up in seven prefectures (Hokkaido, Aomori, Yamagata, Gunma, Aichi, Shiga and Kagawa) | Observational study | 2005 |
| Shiota[34] | 5 550 | Patients of Oita University Hospital, Oita, Japan | Observational study | 2009 |

Pharmaceutical) from November 2000 to December 2015 were as well obtained.

To determine the trend in the number of *H. pylori*-infected individuals, previously published Japanese studies were used (table 1).

### Study design

This study is a retrospective observational study and simulation study based on the health insurance claims data, product sales data and relevant published studies. The steps were as follows: first, the number of individuals who received the eradication therapy and those who had successful eradication were estimated based on the analyses of the health insurance claims databases and product sales data; second, the trend in the number of *H. pylori*-infected individuals was determined from previously published studies; third, the prevalence rate and trend of *H. pylori* infection were estimated and forecasted from the results of the first and second step. Finally, to fully evaluate the impact of the policy changes, a simulation was made considering effects which likely would have occurred without the insurance policy changes in 2000 and 2013.

### Patient identification and analysis

In the JMDC and MDV databases, the individuals who received triple therapy, either the primary eradication package (such as Lansap and other packaged products) or the combination of amoxicillin, clarithromycin and either proton-pump inhibitors or potassium-competitive acid blockers (P-CABs) (all prescribed within the same month), were defined as individuals with primary eradication of *H. pylori*. The drugs used for the therapy were defined by product name in JMDC database and remuneration code in MDV database. The examination was defined by remuneration code, and the diagnosis for those who had eradicated was defined by name of diagnosis in both databases. To estimate the number of individuals who received primary eradication therapy in the nation, the following were calculated in these databases:

1. Percentage of individuals who used Lansap for primary eradication therapy to all individuals who received the primary eradication.
2. Percentage of individuals who received UBIT for the *H. pylori* test after eradication to all individuals who took the *H. pylori* test.

To calculate the number of individuals who achieved successful eradication, the following were assumed (figure 1):

▶ The primary and secondary success rates of eradication for this study were presumed to be 75% and 90%, respectively, based on previous studies in Japan.[25 27] Secondary eradication was premised to be performed for all those who failed primary eradication. Therefore, the success rate of the eradications was estimated to be 98% of the primary eradication, and the percentage was used as the success rate in this study.

▶ New infection in adulthood was reported to be rare[28] and reinfection per year after the eradication therapy in Japan is reported to be approximately 1%[29 30]; however, it was assumed to be 0% in this study.

The nationwide number of individuals who received primary eradication was estimated based on the above points 1. and 2. with sales data of Lansap and UBIT as follows:

▶ The monthly number of individuals who received primary eradication from January 2010 was calculated as the mean of four estimates (Lansap-base from MDV and JMDV, and UBIT-base from MDV and JMDC).

▶ The monthly number of individuals who received primary eradication from January 2006 to December 2009 was calculated as the mean of two estimates from the JMDC database (Lansap-base and UBIT-base).

▶ The monthly number of individuals who received primary eradication from November 2000 to December 2005 was extrapolated using the sales number of UBIT in each month and the UBIT share

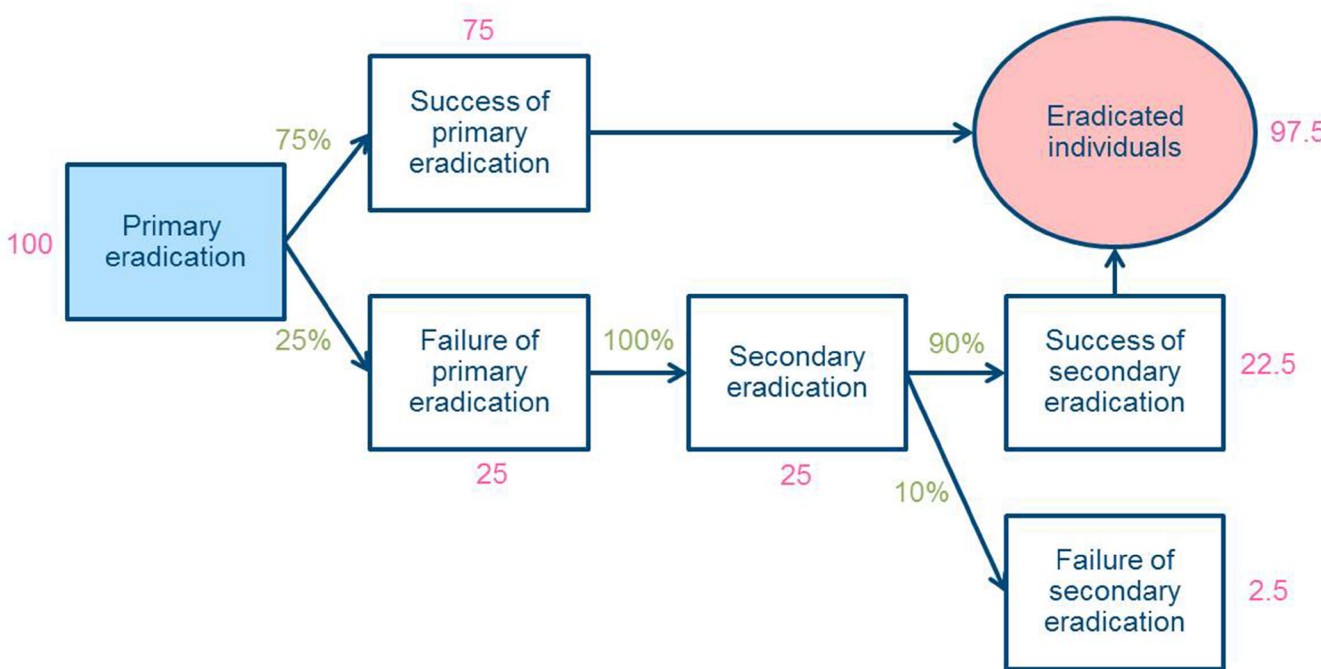

**Figure 1** The model used to calculate the number of individuals with successful *Helicobacter pylori* eradication.

rate in 2006 on the assumption that the share rate in this period was the same as that in 2006.

The formulae used are as follows:
► (Successful eradication number)$_{YM}$ = (Primary successful eradication number)$_{YM}$ + (Secondary successful eradication number)$_{YM}$
► (Primary successful eradication number)$_{YM}$ = (Primary eradication number)$_{YM}$ × (Primary eradication success rate, 75%)
► (Secondary successful eradication number)$_{YM}$= ((Primary eradication number)$_{YM}$ − (Primary successful eradication number)$_{YM}$) × (Secondary eradication success rate, 90%)
► (Primary eradication number)$_{YM}$ = ((Primary eradication number with Lansap)$_{YM,JMDC}$ + (Primary eradication number with Lansap)$_{YM,MDV}$ + (Primary eradication number with UBIT)$_{YM,JMDC}$ + (Primary eradication number with UBIT)$_{YM,MDV}$)/4
► (Primary eradication number with Lansap)$_{YM,database}$ = (Sales number of Lansap)$_{YM}$ / (Share of Lansap)$_{YM,database}$
► (Primary eradication number with UBIT)$_{YM,database}$ = (Examination number)$_{YM,database}$ × (Ratio of primary eradication to total examination)$_{YM,database}$
► (Examination number)$_{YM,database}$ = (Sales number of UBIT)$_{YM}$ / (Share of UBIT)$_{YM,database}$ where YM is year month and database = JMDC or MDV database.

The number of infected individuals and prevalence rate of infection were estimated based on the previous Japanese studies shown in table 1. The number of infected individuals until March 2013 was estimated from previous studies[27 31–34] and vital statistics in Japan conducted by Ministry of Health, Labour and Welfare.[35] An exponential decay approximation curve was calculated based on

the results of the previous studies until 2000. After 2000, the estimated mean monthly number of individuals who achieved successful eradication from January 2001 to March 2013 was taken into account. The number of infected individuals from April 2013 was estimated using the estimated mean monthly number of individuals who achieved successful eradication from April 2013 to December 2015, giving consideration to the decrease in the number of infected individuals due to death. It was calculated as follows:

$$
\text{(Infection number)}_{CY}
= \frac{\sum_{DOBYR} \text{National Population}_{DOBYR,\ CY} \times \text{Prevalence Rate}_{DOBYR,\ CY}}{\sum_{DOBYR} \text{National Population}_{DOBYR,\ CY}}
$$

where (Prevalence rate)$_{DOBYR,CY}$ is the prevalence rate of *H. pylori* infection by birth year in each observation year calculated in this study, and (National population)$_{DOBYR,CY}$ is the population by birth year in each observation year, where DOBYR is birth year by 5 years and CY is the calendar year of observation of each study. An exponential parameter that minimises the sum of squared distances from (Infection number)$_{CY}$ was calculated (least squares method), assuming the decrease of the number of *H. pylori* infected individuals by any reason other than eradication (ie, ageing) followed an exponential function and the decrease of that by eradication was constant from 2001 to 2013.

Assuming that *H. pylori* infection would be decreasing exponentially after 2015 (by consideration of the natural decrease due to the death of older infected individuals), a simulation was performed using the number of infected individuals obtained in the previous section and the population forecast from the National Institute of Population and Social Security Research[36] to predict the number of

infected individuals in the future. The prevalence rate of infection was also simulated for the case of no policy change regarding insurance coverage in 2000 and 2013. This was simulated as follows:

1. The number of *H. pylori*-infected individuals calculated in the previous section was taken to be (Infection number)$_{CY}$ before 2013.

2. The number of *H. pylori*-infected individuals from 2013 to 2015 was assumed to be decreased from (Infection number)$_{CY}$ before 2013, based on the estimated number of individuals who achieved successful eradication by the analysis of the JMDC and MDV databases and sales data of drugs.

3. If case 0, (Prevalence rate)$_{CY,case}$ = (Infection number)$_{CY}$ / (National population)$_{CY}$, in 1975≤CY≤2015 or (Prevalence rate)$_{CY,case}$ / (Prevalence rate)$_{CY-1,case}$ = (Prevalence rate)$_{CY-1,case}$ / (Prevalence rate)$_{CY-2,case}$, in 2016≤CY.

4. If case 1, (Prevalence rate)$_{CY,case}$ = (Infection number)$_{CY}$ / (National population)$_{CY}$, in 1975≤CY≤2000 or (Prevalence rate)$_{CY,case}$ / (Prevalence rate)$_{CY-1,case}$ = (Prevalence rate)$_{CY-1,case}$ / (Prevalence rate)$_{CY-2,case}$, in 2001≤CY.

5. If case 2, (Prevalence rate)$_{CY,case}$ = (Infection number)$_{CY}$ / (National population)$_{CY}$, in 1975≤CY≤2012 or (Prevalence rate)$_{CY,case}$ / (Prevalence rate)$_{CY-1,case}$ = (Prevalence rate)$_{CY-1,case}$ / (Prevalence rate)$_{CY-2,case}$, in 2013≤CY.

Case 0 is with the current policy; case 1 or case 2 represents the case in which policy change had not occurred in 2000 or 2013.

Statistical analysis was carried out using Excel 2010 (Microsoft) and SAS V.9.4 (SAS Institute).

## RESULTS
### Patient characteristics
The total number of individuals who received primary eradication was 81 119 (mean age 36.8 years, males 61%) from the JMDC database and 1 70 993 (mean age 60.6 years, males 57%) from the MDV database. The characteristics for each year are shown in online supplementary appendix, table S1.).

### Trend in the number of individuals with the primary eradication therapy and successful eradication
The difference among the four (Lansap-base from MDV and JMDV, and UBIT-base from MDV and JMDC) estimated numbers of individuals who received primary eradication was confirmed to be minimal after 2010 (see online supplementary appendix, figure S1). The nationwide number of individuals who had successful eradication (both first line and second line) was estimated as shown in figure 2. The number was approximately 650 000 per year between 2001 and 2012, which has reached a steady state of approximately 700 000 per year after an increase in 2006. However, there was a slight decrease in 2011. It markedly increased to 1 380 000 in 2013, which is more

than double the number observed in 2012. In the diagnoses for those who had successful eradication treatment (figure 2), gastritis accounted for more than half of the diagnoses since 2013. The average number of individuals who received successful eradication treatment up to March 2013 was 54 000 per month, whereas it was 124 000 per month after March 2013. The cumulative total of individuals who received successful eradication treatment was more than 10 million up to September 2014.

### Trend in the number of infected individuals and the prevalence rate of infection
Figure 3A illustrates the prevalence rate of *H. pylori* infection by birth year from previous studies. This shows higher prevalence rates of infection in the cohorts with earlier birth years. Also, there was a tendency for the difference of prevalence rates of infection among studies to be larger in those with an earlier birth year, and the rate was lower in later observations. The overall estimated prevalence rate of infection was lower in later years (figure 3B). The lines were fitted after taking the effect of insurance policy change in November 2011 into account, shown in figure 3B.

### Trend and prediction of *H. pylori* infection and the effect of insurance policy changes
The pattern of the number of infected individuals from 2016 in Japan was predicted (blue broken line in figure 4) based on the trend in the number of infected individuals derived from the results above and the population forecast.[35] The patterns, in the case without policy changes in insurance coverage in 2000 and 2013, were also simulated (red and green broken lines in figure 4). The simulation showed that the number of infected individuals would decrease to 16 200 000 individuals in 2030, or 14% of the population, and further decrease to 5% of the population in 2050. These figures would have been 28% and 21% in 2030 if the policy changes had not occurred in 2000 and 2013, respectively.

## DISCUSSION
This study described the status of eradication therapy and trend of *H. pylori* infection using large insurance claims databases that reflected actual clinical practice at the national level in Japan. The analysis showed that the prevalence rate of *H. pylori* infection has decreased after the approval to include eradication therapy in the insurance policy in 2000. The number of successful eradications more than doubled immediately after insurance coverage for *H. pylori* eradication therapy was expanded in 2013. The simulation indicated that the prevalence rate of *H. pylori* infection would decrease and reach approximately 14% in 2030 and 5.4% in 2050.

Although it is difficult to compare the prevalence rate of infection among studies due to time and sample difference, it is worthwhile to compare the rate with other countries. The prevalence rate of *H. pylori* infection varies markedly in different countries; in general, it is higher in

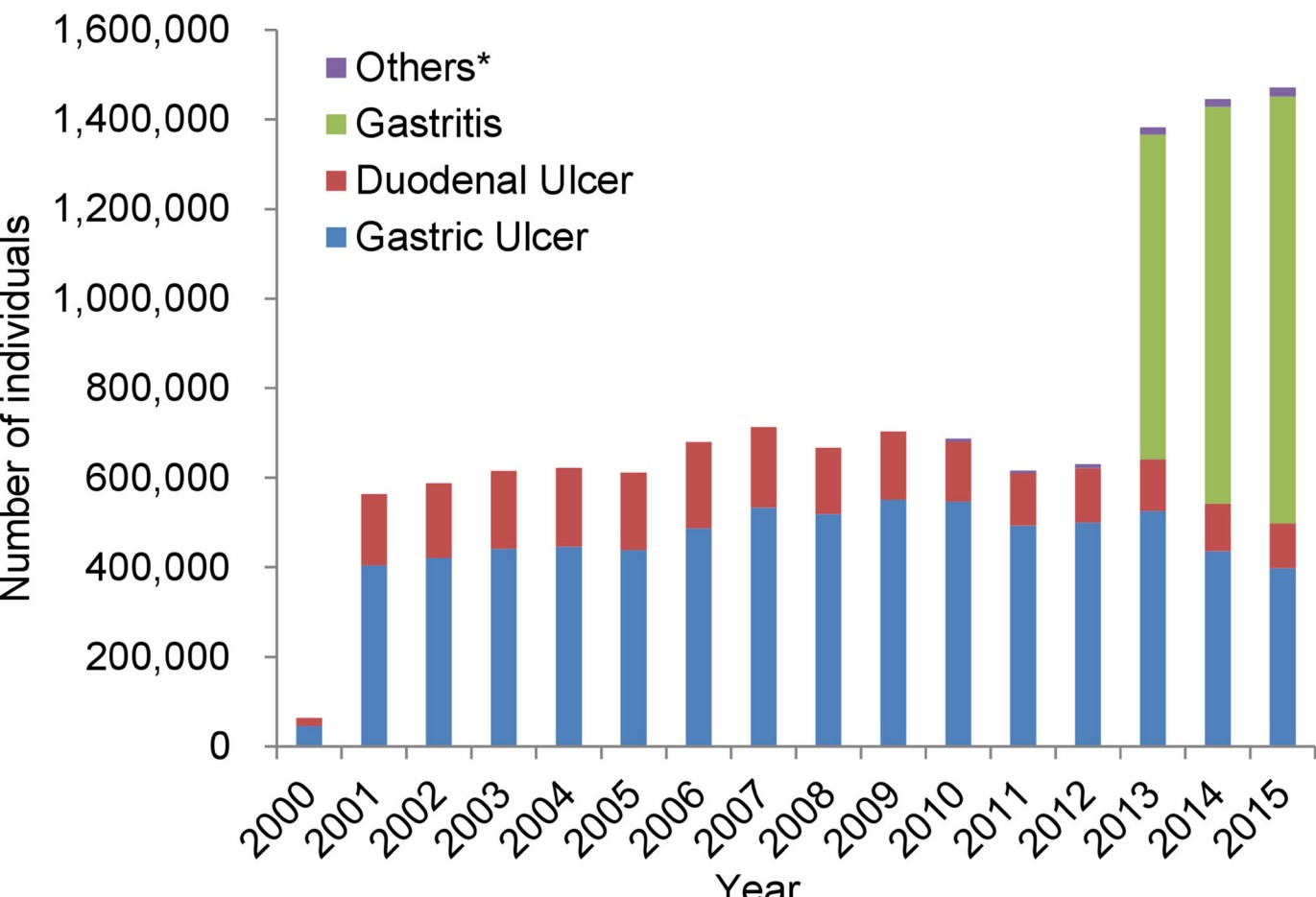

**Figure 2** Annual number of individuals who had successful *Helicobacter pylori* eradication (both first line and second line). *Others: gastric mucosa associated lymphoid tissue lymphoma, idiopathic thrombocytopenic purpura and postendoscopic resection of early gastric cancer.

developing countries and lower in developed countries.[37] The prevalence rate of infection was reported to be 92% in Bangladesh,[38] 75% in Vietnam,[39] 41%–72% in China[40] and 54%–60% in Korea,[41 42] whereas it was 15%–22% in Australia[43 44] and 8%–27% in the USA.[45 46] Although the estimate of the prevalence rate of *H. pylori* infection in our study was 43% in Japan in 2000, based on our simulation, the prevalence rate of infection in Japan in 2030 with expanded insurance coverage would be almost the same as the Australian rate (15%) and it would reach the North American level (8%–27%) by approximately 2050.

The estimate from the previous studies indicated a higher prevalence rate of infection in older cohorts, which can be explained by environmental factors like poor sanitation.[47 48] It has also been suggested that nowadays *H. pylori* infection occurs in childhood in Japan.[49 50] As a result, the number of infections would have naturally decreased even without eradication therapy due to the death of older infected individuals. However, our simulation showed that the prevalence rate of infection would have been higher than 14% and 7% in 2030 if the policy changes had not occurred in 2000 and 2013, respectively. It was, therefore, evident that those insurance policy changes had contributed to the reduction in the

prevalence of *H. pylori* infection. Japan has established a universal health insurance coverage system, which means that by law all Japanese residents are entitled to health insurance coverage for medical treatments. This system has a great impact on the dissemination of medical treatments; consequently, the use of eradication therapy is highly influenced by the presence of the health insurance and coverage for diseases.

The results of this study could be a good indicator for the implementation of insurance coverage for eradication of *H. pylori* in countries where *H. pylori* is prevalent, especially the East Asian countries. Countries and regions, such as Korea, China and Taiwan, have been conducting clinical trials of *H. pylori* eradication; however, they have not yet established any policy for *H. pylori* eradication. This study is likely to be used as one of the references in considering effect on policy change in such countries. This study might also be important in providing direction for future research in Japan. In 2016, the revised edition of the Japanese Guideline for Diagnosis and Treatment for *H. pylori* Infection was published following that of 2009. In the latest guideline, the expansion of insurance coverage for the treatment for *H. pylori* gastritis in 2013 is described.[51] Also, a regimen with P-CAB-based

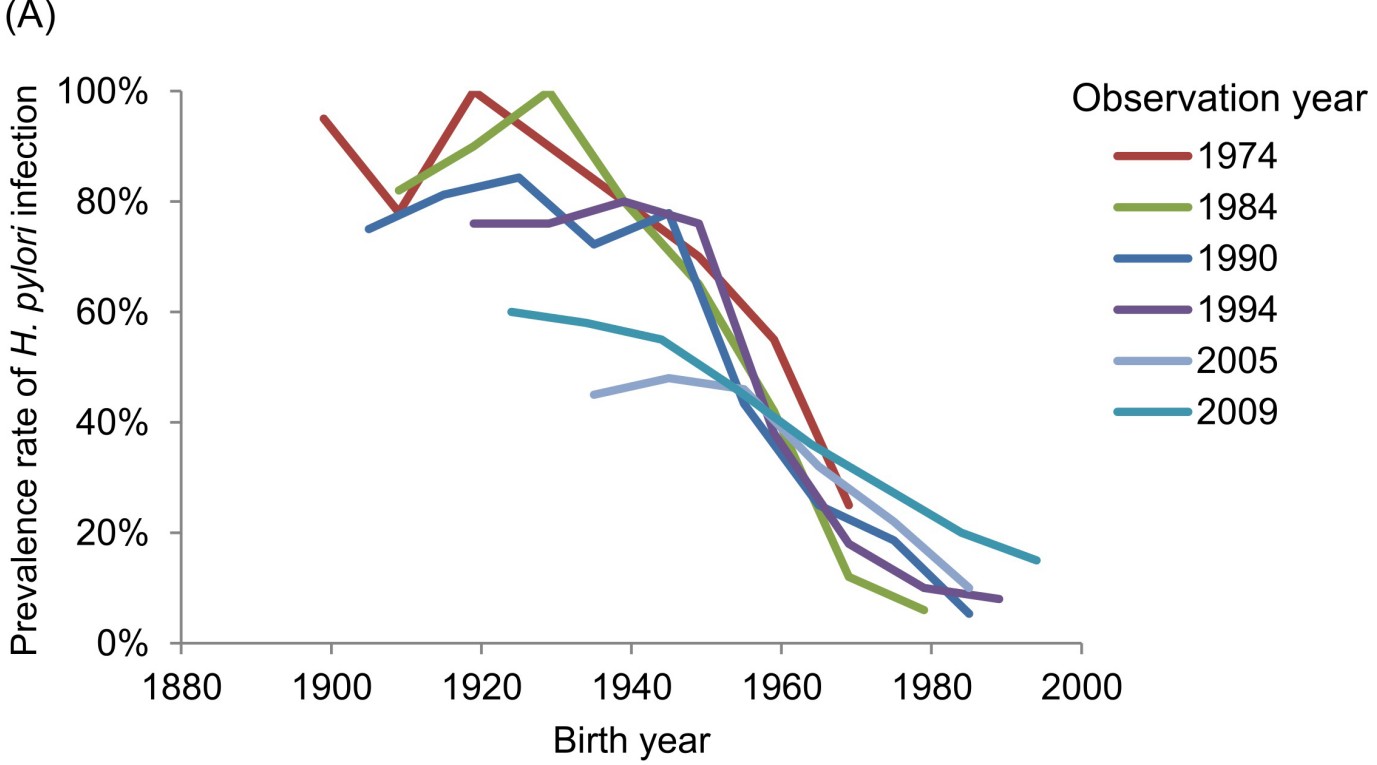

**Figure 3** Trends in (A) the percentages of *Helicobacter pylori* infection by birth year based on previous studies and (B) the nationwide number of infected individuals as estimated based on the previous studies. *Data from the study by Watabe *et al*[32] were excluded in (A) as that study divided the age group into two age groups: below and above age 60 years.

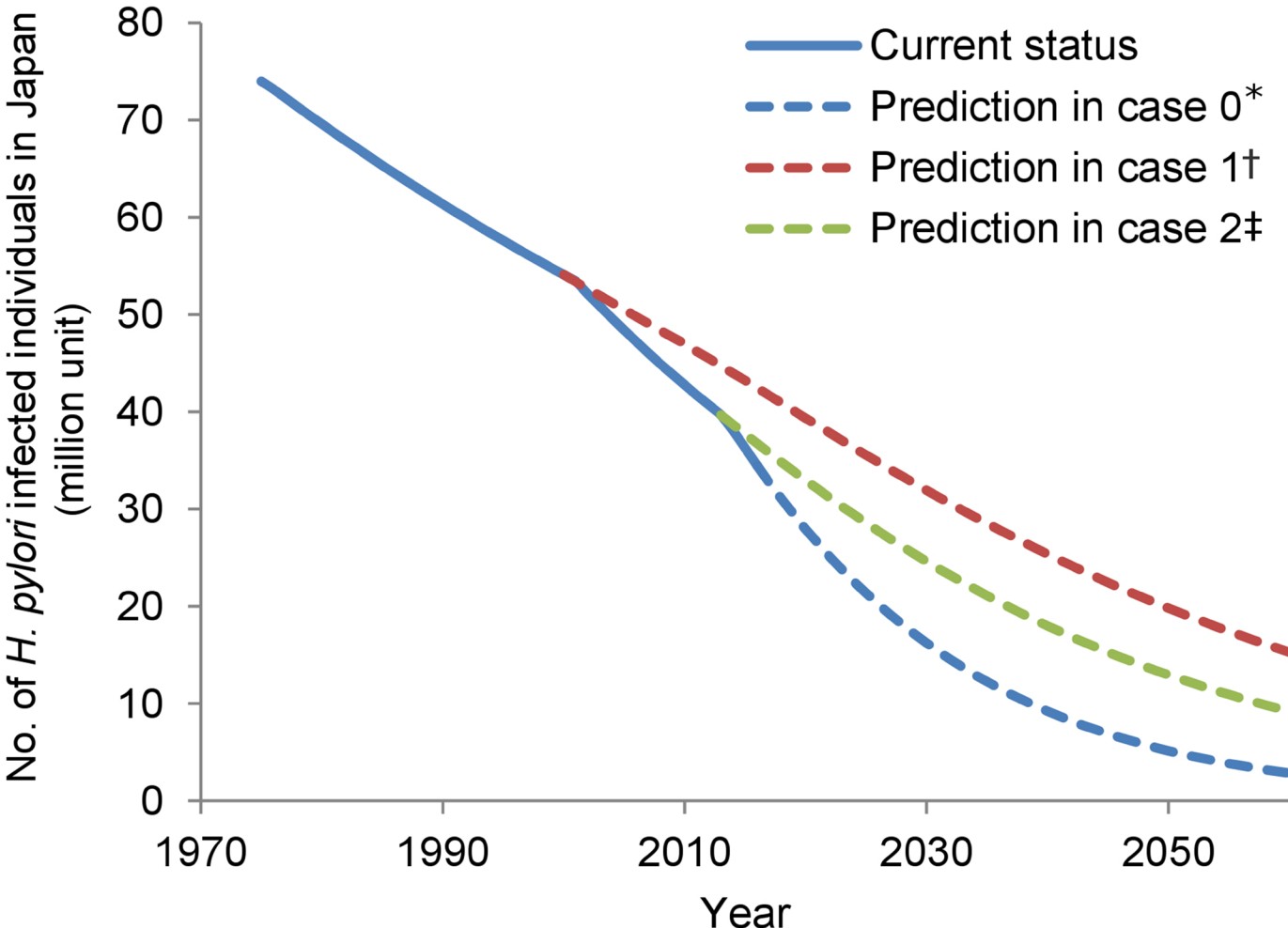

**Figure 4** Trend and prediction of the number of *Helicobacter pylori*-infected individuals. *Case 0, in current policy; †case 1, if policy change had not occurred in 2000; ‡case 2, if policy change had not occurred in 2013.

triple therapy, which is newly described in the guideline, demonstrated a high eradication rate compared with the conventional proton-pump inhibitor-based triple therapy.[51 52] It seems that the prevalence rate of infection could be further reduced in Japan.

### Comparison with other studies

There have been some community-level studies investigating the status of eradication in clinical practices in Japan.[22 23] Nevertheless, to our knowledge, there was no study describing the status of eradication therapy at the national level. Asaka *et al* have reported that the insurance policy change could increase the number of eradications, reduce the number of infected individuals and decrease mortality from gastric cancer.[24 25] They estimated that the number of deaths from gastric cancer would reach 60 000 in 2020 without any countermeasures, whereas it would be half if 50% of infected individuals receive eradication therapy. However, this assumption was not based on actual observations. Our study, using real-world data reflecting actual medical practice, showed a rapid increase in patients receiving *H. pylori* eradication, after the insurance policy change in 2013.

### Strengths and limitations of this study

Here, a national-level evaluation using real-world data, allowed us to analyse the impact of insurance policy expansion for *H. pylori* eradication therapy in a quantitative manner. Furthermore, robust and reliable results were obtained from combinations of large-scale insurance claims databases and sales data of the most commonly used eradication treatments and test kits.

However, this study has several limitations. First, the success rate of eradication was obtained from previous studies[25 27]; thus, the rate might be different from clinical practice, including the possibility that it may be estimated higher than the actual rate considering the effect of the increase in bacteria resistant to antibiotics. Nevertheless, the success rate in this study is believed to be close to the actual rate. Second, the health insurance claims databases have potential biases. The information available was limited for those older than 65 years in the JMDC database because it consisted of information on employed individuals and their family members. In addition, those who were self-employed and employees in small-to-medium sized enterprises were not included.

The information in the MDV database was obtained from the hospital using DPC/PDPS. Indeed, the mean ages of individuals who received primary eradication in both databases were different. However, the product share rates calculated in both databases were very similar after 2010, and the estimates of the number of individuals with the primary eradication were almost identical in both databases. Therefore, the impact of the different age distribution was believed to be minimal, and the estimates are believed to be accurate.

Despite these limitations, this study used the information from reliable clinical studies and large databases covering a significant number of Japanese citizens. Consequently, it is believed that the information presented here reflects the clinical status in Japan.

### Conclusion and policy implications

This study described and forecasted the trend of *H. pylori* eradication therapy and assessed the impact of insurance policy change on the prevalence rate of *H. pylori* infection. It has demonstrated that the policy change was associated with a reduction in the prevalence rate of *H. pylori* infection in Japan. Furthermore, it is expected to lead to a reduction in the incidence of gastric cancer. Adaptation of a similar nationwide health insurance coverage plan for *H. pylori* eradication by other high risk countries and regions may reduce the prevalence of *H. pylori* infection in the short and medium terms and may also have the possibility to have a positive effect on the incidence of *H. pylori*-related conditions, including gastric cancer, in the future.

**Acknowledgements** The authors thank Takeda's collaborator, Otsuka Pharmaceutical, for providing sales data of urea breath test diagnostic kits (UBIT). The authors thank Kosuke Iwasaki and Tomomi Takeshima, employees of Milliman, for analysing the database, and writing and editorial support, respectively, funded by Takeda Pharmaceutical.

**Contributors** SH, KS and KK contributed to the concept and design of the study. SH contributed to acquisition of data. SH contributed to the analysis. SH, KS and ST contributed to interpretation of data. SH, KS, ST and KK contributed to the writing of the manuscript and critical revision of the manuscript. All authors approved the final version of the manuscript. KK is the guarantor of the article.

**Funding** This study was sponsored by Takeda Pharmaceutical.

**Competing interests** SH is an employee of Takeda Pharmaceutical. KS received research grants from Eisai, Daiichi Sankyo Pharma and Takeda Pharmaceutical. He also received lecture fees from Astellas Pharma, Fujifilm and Takeda Pharmaceutical. ST has no personal interests to declare. KK received research funds from Dainippon Sumitomo Pharma, Olympus, Stella Pharma, Medical Platform, Novartis Pharmaceutical, Bayer and Maruho; honorarium from Astellas, Daiichi Sankyo Pharma, Taisho Pharmaceutical, Eisai, Novartis Pharmaceutical, Mitsubishi Tanabe Pharma, Takeda Pharmaceutical and Sanofi; consulting fees from Olympus, Kyowa Hakko Kirin, Kaken Pharmaceutical and Otsuka Pharmaceutical. There are no patents, products in development or marketed products to declare, relevant to those companies.

**Patient consent** Detail has been removed from this case description/these case descriptions to ensure anonymity. The editors and reviewers have seen the detailed information available and are satisfied that the information backs up the case the authors are making.

**Ethics approval** This study was approved by Ethic Committee, Kyoto University Graduate School and Faculty of Medicine (R0126-1). This study was exempted from obtaining individual informed consent based on Ethical Guidelines for Medical and Health Research Involving Human Subjects by Ministry of Education, Culture, Sports, Science and Technology, and Ministry of Health, Labour and Welfare.

**Provenance and peer review** Not commissioned; externally peer reviewed.

**Data sharing statement** No additional data are available.

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
