## [Reviewer comments · BMJ Open]

ARTICLE DETAILS

TITLE (PROVISIONAL)	Impact of health insurance coverage for Helicobacter pylori gastritis on the trends in eradication therapy in Japan: retrospective observational study and simulation study based on real-world data.
AUTHORS	Hiroi, Shinzo; Sugano, Kentaro; Tanaka, Shiro; Kawakami, Koji;

VERSION 1 - REVIEW

REVIEWER	Tao Wang Otsuka Pharmaceutical Development & Commercialization, Inc. United States of America
REVIEW RETURNED	23-Jan-2017

GENERAL COMMENTS	Summary: The authors reported their estimates on the impact national policy changes on the prevalence of H. pylori infection in Japan. Through retrospective analyses of two national health insurance claim databases, the authors showed that updating national policies in treating H. pylori infection resulted in significant increase of use of H. pylori eradication therapy. A trend analysis estimated that the 2 national policy updates lead to significant reduction of number of H. pylori infected individuals in the future beyond normal attrition of infected individuals based on birth years. General Comments: Two outcome variables were used in the retrospective analyses of claim databases: number of individuals who received primary eradication and number of primary eradicated individuals. The nationwide number of individuals who received primary eradication was estimated based on the sales numbers of Lansap® (a clarithromycin-based triple therapy) and UBIT® (a urea breath test). The nationwide number of primary eradicated individuals was estimated from the nationwide number of individuals who received primary eradication (as calculated above) and the success rates of primary and secondary eradication therapy (as reported in the literature). The authors should address the following in revision: 1. It is unclear to this reviewer why the sales number of Lansap itself is not sufficient to estimate the nationwide number of individuals who received primary eradication for H. pylori infection and why the sales number of UBIT is needed. UBIT, as diagnostic test, may be used in individuals suspected of H. pylori infection. Only those who tested positive (presumably approximately 43% based on the estimate in Japan for year 2000) would receive primary eradication therapy. Including sales number of UBIT to estimate the number of
---

individuals who received primary eradication may result in overestimation.

2. In the view of this reviewer, the author should only use the number of individuals who received primary eradication to describe the results of the claim database analyses. Using compounded estimated outcome variables as an outcome variable can introduce unpredictable variability of the data and therefore reduce the credibility of the data. It is the recommendation from this reviewer to use the number of individuals who received primary eradication as the outcome variable to report the claim database analyses results. The other derived outcome variable, estimated number of primary eradicated individuals, should only be used in the trend analysis to demonstrate the effects of H. pylori eradication on future prevalence of H. pylori in Japan.

Specific Comments:

1. Page 2, Abstract/Methods and Page 7, Data Sources, Paragraph 1: Health insurances are intended for a general population, not just for patients. Therefore, change the description of the two databases as follows: JMDC covered 1.6 million lives and MDV covered 10.5 million lives. Use the term patients only when appropriate (e.g., those who received eradication therapy).

2. Page 8, Data Sources, Paragraph 2: The current manuscript estimated the nationwide number of infected individuals from product sales data for the most common eradication medicine and test kit for H. pylori infection. Please explain the rationale and why ICD-10 coding was not used. This reviewer noted that diagnostic information in these two databases is based on ICD-10 codes as described by the authors in the preceding paragraph of the manuscript.

3. Page 8, Data Sources, Table 1: Instead of listing the article titles which can be found in the references, the authors should provide a brief description of the data from the references (e.g., number of subjects, study population, study design, prevalence, etc.) so that the readers can get the sense of the data without going to the references.

4. Page 9, Patient Identification and Statistical Analysis: This study only involves descriptive analysis. Therefore, remove "statistical" from the section title.

5. Page 16, Discussion, Paragraph 2: The authors cited the Sonnenberg's 2010 article as the source of US H. pylori prevalence number. Please add a more frequently cited source of US H. pylori prevalence data (the NHANES) in an article by Cardenas (Cardenas VM, Mulla ZD, Ortiz M and Graham DY. Am J Epidemiol 2005;163:127-134). So change the description to "8% - 27%".

6. Page 17, Discussion, Paragraph 4: Change the following sentence "Countries such as Korea, China, and Taiwan, have been conducting clinical trials of H. pylori eradication" to "Countries and regions such as Korea, China, and Taiwan, have been conducting clinical trials of H. pylori eradication".

7. Page 18, Strengths and Limitations of This Study, Paragraph 2:

	Please add discussion of increasing antibiotic resistance when discussing the first limitation of this analysis. 8. Page 19, Conclusion and Policy Implications: Please delete “so the data in this study may be cited in future studies such as a prediction of gastric cancer mortality”. Whether or not this article will be cited is beyond the scope of this paper.
--	--

REVIEWER	Prof. Lyudmila Boyanova, MD, PhD, DMSc Department of Medical microbiology, Medical University of Sofia, Sofia, Bulgaria
REVIEW RETURNED	31-Jan-2017

GENERAL COMMENTS	The topic of the present manuscript is interesting and emphasizes the impact of Insurance policy expansion for H.pylori eradication therapy and the real and predicted prevalence of the chronic infection. However, there are some additions and changes to be made in order to improve the manuscript. Page 2: Abstract Data about the secondary eradication should be mentioned in the abstract. Study design Page 10, lines 26-27: How was the success rate of the eradication estimated to be 98% of the primary eradication? The usual percentages are much lower according to the literature worldwide. Page 10: Lines 19-23 and 27-28: As for the triple combination including a proton pump inhibitor, amoxicillin, and clarithromycin, the success of eradication has been decreasing over years. Recent data about the eradication rates using clarithromycin-based triple therapy in Japan should be given and discussed. Reference 27 was published >20 years ago, so the eradication success of the triple therapy should be considered based on more data and recent publications. The success of the clarithromycin-based triple therapy depends on clarithromycin resistance rates in the country or the region. What is the primary and what is the post-treatment H. pylori resistance rate in Japan? What were the doses of the antibiotic used and the treatment duration (e.g., 7 or 10 or 14 days)? These are factors that influence the eradication success. Page 10, Lines 23-24: What regimens were used for second-line therapy? Patient identification and statistical analysis Page 10, Lines 36-37: The patients included should be described better. What was the mean age (ranges) of the patients? What were the sex and the proportion of ulcer and non-ulcer diseases? Scanty information is available for this age group. Additional information should be provided. However, the age-cohort phenomenon in H. pylori infection prevalence is well known. Younger birth cohorts usually have a lower H. pylori prevalence
--

	compared with older birth cohorts at the same age. Discussion How many patients (%) needed second-line therapy? Can the authors recommend other eradication regimens to obtain higher eradication rates? References: About 40% of the references were published >10 years ago. Most of them can be replaced by newer data and references.
--	--

REVIEWER	Irene Wong The University of Hong Kong, Hong Kong SAR, China
REVIEW RETURNED	13-Feb-2017

GENERAL COMMENTS	1) It would be beneficial to describe and summarize briefly the effectiveness of eradication therapy in terms of gastric cancer-related outcomes in the Introduction. This would illustrate the usefulness of studying the trends in eradication therapy. 2) Please add descriptions about health care system, user payment methods and access of care in Japan as the background information. Moreover, would it be possible that the insurance coverage affect the equity and access to care in Japan, and therefore affect the related trends? What do you think? It would be beneficial to expand the Discussion on this issue. 3) Please evaluate the representativeness and validity of data of insurance coverage and eradication. Moreover, how might the coverage of data used potentially affect validity of the study findings? Please discuss. 4) In the Methods, to facilitate one's understanding, please describe briefly the rationales of the formula and related calculations. In particular, with respect to the formula 3 to formula 5 on page 13, they are the key methodologies to estimate the effects of insurance coverage on the trends of eradication therapy. It appears to be an extrapolation method based on very simplistic assumptions. This would potentially affect the validity of study findings. 5) Any potential under-reporting issues? Please address and discuss in the Methods and Discussion. 6) Insurance coverage may not be the only contributing factor on the trends in eradication therapy. It might not be necessary that more insurance coverage would lead to successful eradications. It could be also due to other factors such as eradication effectiveness in Japanese, health care access, cultural health education and health care seeking behavior pattern in Japan. This requires more discussion.
--

REVIEWER	Jørgen T Lauridsen University of Southern Denmark
REVIEW RETURNED	20-Feb-2017

GENERAL COMMENTS	The study is professionally performed in all respects. Only a few minor concerns for your consideration:  - The methods section presents the formulas in a somewhat lengthy and text form. However, I agree that this may be easier to understand for a non-technical reader than a more compressed
---

	mathematical form. Anyway, I kindly ask you to give this a second thought. - The reference list appears comprehensive. However, there seem to be a slight underrepresentation of very new (2015-16) international references. Please give it a last try with a literature search for new papers. - You claim to use SAS Version 9.2. Indeed, I have used 9.3 for years and is now using 9.4. Please check that your information is correct.
--	--

VERSION 1 – AUTHOR RESPONSE

To Reviewer 1

Reviewer Name: Tao Wang

Institution and Country: Otsuka Pharmaceutical Development & Commercialization, Inc. United States of America

General comments:

1. It is unclear to this reviewer why the sales number of Lansap itself is not sufficient to estimate the nationwide number of individuals who received primary eradication for H. pylori infection and why the sales number of UBIT is needed. UBIT, as diagnostic test, may be used in individuals suspected of H. pylori infection. Only those who tested positive (presumably approximately 43% based on the estimate in Japan for year 2000) would receive primary eradication therapy. Including sales number of UBIT to estimate the number of individuals who received primary eradication may result in overestimation.

Response: Thank you for insightful comments. We agree that the sales number of Lansap is reliable data to estimate the number of individuals who received primary eradication for H. pylori infection. We also thought that our simulation would be more accurate if we have an estimate from a different model.

We believe that the sales number of UBIT could be another data source if it was adjusted for the ratio of the share of UBIT to the number of primary eradication obtained from our claims database.

2. In the view of this reviewer, the author should only use the number of individuals who received primary eradication to describe the results of the claim database analyses. Using compounded estimated outcome variables as an outcome variable can introduce unpredictable variability of the data and therefore reduce the credibility of the data. It is the recommendation from this reviewer to use the number of individuals who received primary eradication as the outcome variable to report the claim database analyses results. The other derived outcome variable, estimated number of primary eradicated individuals, should only be used in the trend analysis to demonstrate the effects of H. pylori eradication on future prevalence of H. pylori in Japan.

Response: We used a model to show the number of individuals who had successful eradication on the primary therapy. We added the model as Figure 1. Although we can understand your concern about the credibility of the data as you mentioned, estimation of the number of successfully eradicated individuals was one of our objectives in this study. So we described the number of these individuals as one of outcomes with assumption of the success rate of eradication. We also describe the basis of calculation; so that we believe the number is meaningful as an estimate.

Specific Comments:

1. Page 2, Abstract/Methods and Page 7, Data Sources, Paragraph 1: Health insurances are intended for a general population, not just for patients. Therefore, change the description of the two databases as follows: JMDC covered 1.6 million lives and MDV covered 10.5 million lives. Use the term patients only when appropriate (e.g., those who received eradication therapy).

Response: As you pointed out, the word “patients” is not appropriate. Since we basically use the word

“individuals” in other parts, we replaced the word “patients” with “individuals” on pages 2 and 8 in the revised manuscript.

2. Page 8, Data Sources, Paragraph 2: The current manuscript estimated the nationwide number of infected individuals from product sales data for the most common eradication medicine and test kit for *H. pylori* infection. Please explain the rationale and why ICD-10 coding was not used. This reviewer noted that diagnostic information in these two databases is based on ICD-10 codes as described by the authors in the preceding paragraph of the manuscript.

Response: We know *H. pylori* infection is coded by ICD-10 code; however, the code is not always given when someone eradicates. Therefore, we considered that it is accurate to define the individuals who took eradication therapy by prescription with drugs used to standard eradication therapy.

3. Page 8, Data Sources, Table 1: Instead of listing the article titles which can be found in the references, the authors should provide a brief description of the data from the references (e.g., number of subjects, study population, study design, prevalence, etc.) so that the readers can get the sense of the data without going to the references.

Response: Thank you for your useful comment. According to your suggestion, in Table 1 we have removed the titles of studies, and in place, added the number of subjects, study population and study design. We did not add the prevalence because it is already shown in Figure 3a by each birth year in the revised manuscript.

4. Page 9, Patient Identification and Statistical Analysis: This study only involves descriptive analysis. Therefore, remove “statistical” from the section title.

Response: We removed “statistical” from the section title as you advised on page 9 in the revised manuscript.

5. Page 16, Discussion, Paragraph 2: The authors cited the Sonnenberg’s 2010 article as the source of US *H. pylori* prevalence number. Please add a more frequently cited source of US *H. pylori* prevalence data (the NHANES) in an article by Cardenas (Cardenas VM, Mulla ZD, Ortiz M and Graham DY. *Am J Epidemiol* 2005;163:127-134). So change the description to “8% - 27%”.

Response: Thank you for your suggestion. We cited the article (as reference #46) and modified the prevalence of *H. pylori* in US as 8-27% in 2nd paragraph in Discussion session on page 16 in the revised manuscript.

6. Page 17, Discussion, Paragraph 4: Change the following sentence “Countries such as Korea, China, and Taiwan, have been conducting clinical trials of *H. pylori* eradication” to “Countries and regions such as Korea, China, and Taiwan, have been conducting clinical trials of *H. pylori* eradication”.

Response: We modified the sentence as you suggested in 4th paragraph of Discussion session on page 17 in the revised manuscript.

7. Page 18, Strengths and Limitations of This Study, Paragraph 2: Please add discussion of increasing antibiotic resistance when discussing the first limitation of this analysis.

Response: We described the possibility of the effect of an increase in bacteria resistant to antibiotics in the Limitations section on page 19 in the revised manuscript.

8. Page 19, Conclusion and Policy Implications: Please delete “so the data in this study may be cited in future studies such as a prediction of gastric cancer mortality”. Whether or not this article will be cited is beyond the scope of this paper.

Response: We deleted the sentence in the Conclusion and Policy Implications section on page 20 in the revised manuscript as you advised.

To Reviewer 2

Reviewer Name: Prof. Lyudmila Boyanova, MD, PhD, DMSc

Institution and Country: Department of Medical microbiology, Medical University of Sofia, Sofia, Bulgaria

1. Page 2: Abstract

Data about the secondary eradication should be mentioned in the abstract.

Response: Thank you for your comments. We also thought it is better to describe secondary eradication, but it is difficult due to the constraints of the word limit. Our Abstract is 299 words out of the word limit of 300 words. Instead of describing secondary eradication, we added a figure of a model including secondary eradication as Figure 1 to make the entire design, including second eradication clear.

2. Study design

Page 10, lines 26-27: How was the success rate of the eradication estimated to be 98% of the primary eradication? The usual percentages are much lower according to the literature worldwide.

Response: We estimated the success rate of entire eradication to be 98%, which is not only that of primary eradication. We added an estimation model of eradication success rate as Figure 1 including the presumed success rate in the primary and secondary eradication to make it clear.

3. Page 10: Lines 19-23 and 27-28: As for the triple combination including a proton pump inhibitor, amoxicillin, and clarithromycin, the success of eradication has been decreasing over years. Recent data about the eradication rates using clarithromycin-based triple therapy in Japan should be given and discussed. Reference 27 was published >20 years ago, so the eradication success of the triple therapy should be considered based on more data and recent publications.

Response: As you mentioned, we also recognize that the eradication rate has been changing due to the increase in resistance to clarithromycin, etc. In this study, we predicted the number of successfully eradicated individuals for a long period of time, from the past to the future. Although we might need to change the success rate for each time period to calculate the number, we used one success rate to predict it from the past to the future. Actually, we calculated the eradication success rate of *H. pylori* in Japan based on Real World Data, not published yet. As a result, the success rate was confirmed to remain around 80% in recent years; so we consider the eradication success rate in the first eradication of 75% is at least not an overestimate.

4. The success of the clarithromycin-based triple therapy depends on clarithromycin resistance rates in the country or the region. What is the primary and what is the post-treatment *H. pylori* resistance rate in Japan?

Response: In Japan, the resistance rate to clarithromycin has been reported to be 7.1% in 2000 when the insurance coverage for clarithromycin-based triple therapy started. The rate has been increasing; 24.7%, 31.0%, and 38.5% from 2002–2006, 2010–2011, 2013–2014, respectively.

(Reference: Hashinaga et al., Japanese Journal of Helicobacter Research, 2016;17:45-49)

5. What were the doses of the antibiotic used and the treatment duration (e.g., 7 or 10 or 14 days)? These are factors that influence the eradication success.

Response: The dosages of antibiotic in standard treatment are 1500 mg/day for amoxicillin, and 400 or 800 mg/day for clarithromycin. The treatment duration is seven days only. They are regulated by a restriction of insurance coverage. Although the dosages may affect the eradication success rate as you mentioned, in this study we did not examine the relationship between regimen and eradication success rate.

6. Page 10, Lines 23-24: What regimens were used for second-line therapy?

Response: The regimen for the secondary eradication is amoxicillin (1500 mg/day), metronidazole (500 mg/day), and proton pump inhibitor in Japan.

7. Patient identification and statistical analysis

Page 10, Lines 36-37: The patients included should be described better. What was the mean age (ranges) of the patients? What were the sex and the proportion of ulcer and non-ulcer diseases?

Response: We describe the mean age and percentage of the male patients in Supplementary information as Table S1. We added standard deviation as age range to Table S1. The number of successfully eradicated individuals having peptic ulcer (duodenal ulcer and gastric ulcer) is shown in Figure 2 in the revised manuscript.

8. Scanty information is available for this age group. Additional information should be provided. However, the age-cohort phenomenon in H. pylori infection prevalence is well known. Younger birth cohorts usually have a lower H. pylori prevalence compared with older birth cohorts at the same age.
Response: As you mentioned, it is known that the infection rate of H. pylori is different in each age group in Japan as well. We also examined the infection rate of H. pylori by birth year as shown in Figure 3a in the revised manuscript.

Discussion

9. How many patients (%) needed second-line therapy?

Response: We assumed the percentage of individuals who needed the secondary eradication to be 25%, as shown in Figure 1, as well as the 75% as success rate in Patient identification and analysis section of Methods on page 10 in the revised manuscript. In fact, we confirmed that the recent rate of those needing secondary eradication remains around 20% based on a Real World Data analysis in Japan (unpublished data).

10. Can the authors recommend other eradication regimens to obtain higher eradication rates?

Response: In this study, we did not compare the eradication success rate among regimens; therefore, we did not propose any recommendation. We are considering examination of the success rate in each regimen in the future.

References:

11. About 40% of the references were published >10 years ago. Most of them can be replaced by newer data and references.

Response: We searched to find newer studies using PubMed according to your comments. As a result, related to the prevalence of H. pylori infection, we found three newer studies: China (#40, published in 2015), Korea (#42, published in 2013), and Australia (#44, published in 2016), and one study for USA (#46, published in 2006) was recommended by Reviewer 1. Then removed the previous #40, published in 2003.

To Reviewer 3

Reviewer Name: Irene Wong

Institution and Country: The University of Hong Kong, Hong Kong SAR, China

1) It would be beneficial to describe and summarize briefly the effectiveness of eradication therapy in terms of gastric cancer-related outcomes in the Introduction. This would illustrate the usefulness of studying the trends in eradication therapy.

Response: Thank you for your comments. We described that H. pylori can cause gastric inflammation, which can then lead to gastric cancer, and the eradication of H. pylori is considered to be effective to reduce the risk of gastric cancer in the first paragraph of the Introduction section on page 5 in the revised manuscript.

2) Please add descriptions about health care system, user payment methods and access of care in Japan as the background information. Moreover, would it be possible that the insurance coverage affect the equity and access to care in Japan, and therefore affect the related trends? What do you think? It would be beneficial to expand the Discussion on this issue.

Response: Thank you for your suggestion. As you suggested, we added a description about health care system to the 3rd paragraph of the Introduction section on page 6 in the revised manuscript.

3) Please evaluate the representativeness and validity of data of insurance coverage and eradication. Moreover, how might the coverage of data used potentially affect validity of the study findings? Please discuss.

Response: The two databases that we used in this study are representative nationwide databases available in Japan. Also, they are the only databases used by regulatory authorities. As you pointed out, there are concerns regarding representativeness and validity of data; however, the evaluation is difficult due to the availability of database. Thus, we confirmed the robustness of the results by four types of approaches: the combination of two types of insurance databases and two types of sales databases, eradication medicine (Lansap®) and test kit (UBIT®).

4) In the Methods, to facilitate one's understanding, please describe briefly the rationales of the formula and related calculations. In particular, with respect to the formula 3 to formula 5 on page 13, they are the key methodologies to estimate the effects of insurance coverage on the trends of eradication therapy. It appears to be an extrapolation method based on very simplistic assumptions. This would potentially affect the validity of study findings.

Response: We described the methods by both formulas and sentences to make our methods understandable for both technical and non-technical readers. As for formula 3 to formula 5, we consider that it is reasonable to apply extrapolation methods because the transition of number of infected individuals fit with an exponential curve as shown in Figure 3b in the revised manuscript.

5) Any potential under-reporting issues? Please address and discuss in the Methods and Discussion.

Response: All studies that we used to estimate the number of H. pylori infected individuals were observational studies. Therefore, we consider that the possibility of under-reporting is low. Also, we added the study design and population of these studies to Table 1.

6) Insurance coverage may not be the only contributing factor on the trends in eradication therapy. It might not be necessary that more insurance coverage would lead to successful eradications. It could be also due to other factors such as eradication effectiveness in Japanese, health care access, cultural health education and health care seeking behavior pattern in Japan. This requires more discussion.

Response: We think factors other than insurance coverage relatively consistent throughout the study period. Therefore, the trends in eradication therapy can be attributed to the change in insurance coverage.

Since the health insurance system in Japan is universal coverage, the people are ensured free and equal access to health care. Although cultural health education and health care seeking behavior pattern might also have an effect, as you pointed out, the effect at the time following the insurance coverage change was drastic, as shown in Figure 2 in the revised manuscript. Although H. pylori eradication was recommended for H. pylori-positive gastritis in the third revision of the guideline published in 2010 by the Japanese Helicobacter Research Society (Reference #21), we observed little change in the number of persons who received eradication therapy up to 2013 when insurance coverage started.

Reviewer Name: Jørgen T Lauridsen
Institution and Country: University of Southern Denmark

1. - The methods section presents the formulas in a somewhat lengthy and text form. However, I agree that this may be easier to understand for a non-technical reader than a more compressed mathematical form. Anyway, I kindly ask you to give this a second thought.

Response: Thank you for your suggestive comments. We presented the formulas such that they may be easily understandable for both non-technical readers and technical readers as you mentioned.

2. - The reference list appears comprehensive. However, there seem to be a slight underrepresentation of very new (2015-16) international references. Please give it a last try with a literature search for new papers.

Response: We searched to find newer studies using PubMed according to your comments. As a result, related to the prevalence of H. pylori infection, we found three newer studies: China (#40, published in 2015), Korea (#42, published in 2013), and Australia (#44, published in 2016), and one study for the USA (#46, published in 2006) was recommended by Reviewer 1. Then we removed the previous #40, published in 2003.

3. - You claim to use SAS Version 9.2. Indeed, I have used 9.3 for years and is now using 9.4. Please check that your information is correct.

Response: Thank you for kind comments. After confirmation, we corrected it to Version 9.4.

VERSION 2 – REVIEW

REVIEWER	Tao Wang Otsuka Pharmaceutical Development and Commercialization, Inc. United States of America.
REVIEW RETURNED	27-Apr-2017

GENERAL COMMENTS	This revised manuscript is acceptable for publication and does not need to be reviewed again by me if the authors can make the changes similar to the following: 1. Abstract - Conclusions: "Recent policy change for H. pylori eradication therapy in Japan was associated with a reduction in the prevalence of H. pylori infection in Japan. This is expected to lead to a reduction in the incidence of gastric cancer." 2. Conclusion and policy implications: Revise the last sentence to something like "Adaption of similar nation-wide health insurance coverage for H. pylori eradication by other high risk countries and regions may have the same impact on the prevalence of H. pylori infection and associated conditions."
--

REVIEWER	Prof. Lyudmila Boyanova, MD, PhD, DMSc Department of Medical Microbiology, Medical university of Sofia, Sofia, Bulgaria
REVIEW RETURNED	06-May-2017

GENERAL COMMENTS	The authors have replied to my question and have revised the manuscript. I have no additional questions.
--

REVIEWER	Irene Wong The University of Hong Kong, Hong Kong SAR, China
REVIEW RETURNED	12-May-2017

GENERAL COMMENTS	No added comment.
-------------------

REVIEWER	Jørgen T. Lauridsen University of Southern Denmark, Denmark
REVIEW RETURNED	03-May-2017

GENERAL COMMENTS	The study is concerned with the effect of insurance covery on utilisation of eradication therapy for H. pylori., which in turn is known to cause gastric cancer. Given that this is underdescribed in previous studies, an innovative contribution is aimed at that merits publication. The study is motivated, and relevant literature surveyed. Methods are suitable and well described. Data are extensive and fully utilised to their best. Results are described and sufficiently discussed. Limitations are outlined and appears not to have serious effects on the outcome. The English style is satisfactory. Thus, I recommend publication without reservation.
---

VERSION 2 – AUTHOR RESPONSE

Our response to the comment of Reviewer 1.

To Reviewer: 1

Reviewer Name: Tao Wang

Institution and Country: Otsuka Pharmaceutical Development and Commercialization, Inc. United States of America.

1. Abstract - Conclusions: "Recent policy change for H. pylori eradication therapy in Japan was associated with a reduction in the prevalence of H. pylori infection in Japan. This is expected to lead to a reduction in the incidence of gastric cancer."

Response: Thank you for your comments. As you mentioned, a reduction in the incidence of gastric cancer can be also expected. However, we only examined the influence on the individuals infected with H. pylori; we did not examine the incidence of gastric cancer in this study. Actually, we are currently conducting a study related to the incidence and mortality associated with gastric cancer, so we will discuss this topic in a future paper. Also, we expect that the results from the policy change on the reduction in H. pylori infection in Japan can be useful reference for policy decision-making in other countries, especially those with a high H. pylori infection rate; therefore, we would like to keep this discussion. Regarding the word limit as well, we would like to keep the sentences in the Conclusions of the Abstract.

2. Conclusion and policy implications: Revise the last sentence to something like "Adaption of similar nation-wide health insurance coverage for H. pylori eradication by other high risk countries and regions may have the same impact on the prevalence of H. pylori infection and associated conditions."

Response: Thank you for your useful comments. We have modified the last sentence of the section,

Conclusion and policy implications by giving consideration to your suggestion as follows:

“Adaption of a similar nationwide health insurance coverage plan for H. pylori eradication by other high risk countries and regions may reduce the prevalence of H. pylori infection in the short and medium terms and may also have the possibility to have a positive effect on the incidence of H. pylori-related conditions, including gastric cancer, in the future”.